# Cyclic Testing of Polymer Composites and Textile Cords for Tires

**DOI:** 10.3390/polym15102358

**Published:** 2023-05-18

**Authors:** Jan Krmela, Michal Michna, Zdeněk Růžička, Vladimíra Krmelová, Artem Artyukhov

**Affiliations:** 1Faculty of Mechanical Engineering, Jan Evangelista Purkyně University in Ustí nad Labem, 400 96 Ustí nad Labem, Czech Republic; michal.michna@sps-cl.cz (M.M.); zdenek.ruzicka@centrogordc.cz (Z.R.); 2Faculty of Industrial Technologies in Púchov, Alexander Dubček University of Trenčín, 020 01 Púchov, Slovakia; vladimira.krmelova@tnuni.sk; 3Academic and Research Institute of Business, Economics and Management, Sumy State University, 40007 Sumy, Ukraine; a.artyukhov@pohnp.sumdu.edu.ua

**Keywords:** composite, PA66, cyclic test, temperature, relaxation, true stress, tire

## Abstract

This paper is oriented toward the specific testing of polymer composites and textile PA66 cords used as reinforcement for composites. The aim of the research is to validate the proposed new testing methods for low-cyclic testing of polymer composites and PA66 cords for the characterization of material parameters useful as input data for computational tire simulations. Part of the research is the design of experimental methods for polymer composites and test parameters such as load rate, preload, and other parameters such as strain for the start and stop of cycle steps. The DIN 53835-13 standard is used for the conditions of textile cord during the first five cycles. A cyclic load is carried out at two temperatures of 20 °C and 120 °C. The testing method includes a hold step for 60 s between each loop. The video-extensometer technique is used for testing. The paper evaluated the effect of temperatures on the material properties of PA66 cords. The true stress-strain (elongation) dependences between points for the video-extensometer of the fifth cycle of every cycle loop are the data results from composite tests. The forcestrain dependences between points for the video-extensometer are the data results from tests of the PA66 cord. These dependencies can be used as input material data of textile cords in the computational simulation of tire casings using a custom material model definition. The fourth cycle in every cycle loop of polymer composites can be considered a stable cycle because the change in the maximum true stress between the fourth and fifth cycles is 1.6%. Other results of this research include a relationship between stress and the number of cycle loops as the second-degree polynomial curve for polymer composites and a simple relationship to describe the value of the force at each end of the cycles for a textile cord.

## 1. Introduction

The paper focuses on cyclic experiments of polymer composites with textile reinforcements (called as cords), which are used for automobile transport in the interior and especially in tire manufacturing. The authors have been devoted to tire casings and tires for a long time. Therefore, tire casing materials are used for this research. Tires are one point of contact between the vehicle and the road surface [1], ensuring the transmission of forces and moments [2] and also contributing to the suspension of the vehicle. The tire consists of a tire casing and a rim. An automobile radial tire casing with classical construction [3] consists of rubber with textile and steel cords [4] in a tire crown and carcass, Figure 1. The construction parts applied to the radial tire casings of the passenger car are a steel cord belt, a textile carcass, and a textile overlap belt [5,6]. These construction parts are themselves composites, consisting of one or several plies. The description of the tire casing with details of the microstructures of the reinforcing ply is given in the publication [6]. Information about the cord materials used and the number of plies in the sidewall and under the tread of the tire casing can be found on the tire sidewall. There is one polyester layer in the sidewall of a passenger car tire and four construction layers in the crown of the tire casing, namely [6]: one polyester layer, two steel layers, and one polyamide layer. The different materials applied for the textile reinforcements in the textile carcass and a textile overlap belt are listed in Table 1 and are also determined by the purpose of the shell, e.g., for sports purposes, aramid as a cord material is preferable.

For the overlap belt, polyamide PA66 140 × 1 × 2 is used as a typical construction of textile reinforcement [6]. The density of the textile ply is the number of ends per one meter of width. Density is referred to as EPM. The typical density value is in the range of 1000 m^−1^ to 1200 m^−1^. As a sample, the EPM of the textile overlap belt of the Dunlop 215/40 R17 87V Extra load SS SPORT MAXX tire casing is approximately 1400 m^−1^. The diameter of the cords is 0.44 mm, and the thickness of the overlap belt is 1.1 mm. The main geometric parameters of the construction parts of Dunlop 215/40 R17 tire casings as diameters of cords are shown in Table 2. The parameters of the reinforcements of the Continental 245/40 R18 97Y Extra load ContiSportContact 3 tire casing are in Table 3; the parameters are different. The textile carcass data in Table 2 and Table 3 are important for the preparation of one-layer composite specimens.

Statical tensile tests of composite specimens are important for obtaining knowledge about stiffness characteristics and material parameters. Cyclic loading tests of composite parts of tire casings and textile cords are requested for the verification analysis between experiments and computational simulation of tires and as input data (about cords) to simulations under dynamic loading. Tests of specific polymer composites with an elastomer matrix are not standardized. The paper [7] describes the geometric parameters of single-layer tire casing specimens with different cord-angles (0°, 25°, 45°, 60°, and 90°) for uniaxial tensile tests. Some standards describe the test procedure and testing conditions, the shape, and the geometric parameters of the test specimens for static tensile tests of composites [8,9]. The ASTM D-3039-76 standard [10] describes the width of fiber-resin composite test specimens in the form of strips 25 mm wide and 2–4 mm thick with a working length of 150 mm. Therefore, for cyclic loading tests, the shape of specimens of a single layer with textile cords and elastomer must be designed. For textile materials and fibers, the DIN 53835-13 standard [11] defines conditions for cycle loading for the first five cycles. This standard can be used for PA66 cords. Next, it is possible to rely on knowledge of the literature focused on the specific loading of polymers, e.g., [12].

The aim of the authors’ research is to propose experimental methods for composites and textile cords that simulate the real cycle states of operational loading during tire running. The cyclic loading tests of composites are requested for verification analyses between tests and the computational simulation of tires too. On the basis of the literature search, it can be concluded that there is no similar publicly available publication that deals with the design of methods for low-cycle loading of polymer composites and tire cords with the aim of mathematical interpretation of the results and the application of a video-extensometer.

## 2. Materials and Methods

### 2.1. Material of Composite Specimens

Composite specimens with a rectangular shape consist of nylon PA66 as textile reinforcement cords with a diameter of 0.50 mm, an EPM of 870 m^−1^, and a ply thickness of 1.1 ± 0.05 mm. Matrix is a rubber that is currently used for cap plies of Continental passenger tires. The composite structure represents a part of the tire casing—the textile cap ply of the unspecified tire casing (factory specimens provided by the manufacturer). The specimen parameters are designed with a length of 195 mm, a width of 35 mm (based on the width of the jaws of the tensile test machine). For testing, the initial length between the jaws of the test machine of 100 mm is used. Specimens with a 45° angle of the textile cord are used for cyclic tests. The angle of the cord was chosen purposefully based on the research in [7] and also for fast production of test specimens.

### 2.2. Material of Textile Cord

For textile cord experiments, PA66 textile yarns with construction 470 × 2 are used, which means linear density—fineness 940 dtex. These textile yarns are usually used as a textile cord for tire producers. This single-end cord product name is Kordsa T-728 [13]. Kordsa T-728 textile cord is manufactured by Kordsa Company. Based on ASTM D 1776-08 standards [14], the fineness of the textile-polyamide tire cord must be measured, as well as the tensile tests, for 65 ± 2% relative humidity and 20 ± 2 °C temperature. The real value of the fineness of the textile-conditioned cord is 1003 dtex based on measurement with an analytical laboratory balance. The literature [15], made by A. Yılmaz, from company Kordsa Turkey, used different conditions for the yarn testing: 55 ± 2% relative humidity and 24 ± 2 °C temperature, which is based on the ASTM D 885-07 standard [16] valid for rayon yarns and rayon tire cords.

Kordsa T-728 is a specially designed multifilament PA66 yarn for advanced industry solutions. It has superior mechanical yarn quality with advanced physical properties [13]. T-728 is mainly used in modular Tire Building Technology [17]. It is also widely used in hoses, air springs, V belts, etc. The material parameters of T-728 with 940 dtex are in Table 4.

Literature [15] is a publication available with almost complete information about the Kordsa T-728, there is no more recent publication that contains similar information as [15]. That is why this publication is unique and can be used because this Kordsa T-728 cord is still in use. Some of the material parameters are also listed in the manufacturer’s data [13], but here we find a discrepancy in numbers. If the breaking strength [13] is 8.3 kg, then the breaking force is 81.4 N (the same result as is also written in the literature [15]), but based on the recalculation of tenacity for linear density 940 dtex, we get the tenacity 86.6 cN·dtex^−1^, but in ref. [13], there is 86.2 cN·dtex^−1^ which would correspond in the recalculation for linear density 944 dtex. Considering that in ref. [13], the linear density is 940 dtex, it can be stated that the information in [13] can only be taken as supplementary information, and the primary publication [15] is essential for our research.

### 2.3. Testing Method for Composite Specimens

Polymer composites do not have standardized uniaxial cycling tests. Therefore, part of the research is the design of an experimental method for these composites with specified test parameters such as load rate, preload, and other parameters—strain for the start and stop of cycle steps. For testing, the software Shimadzu TrapeziumX version 1.5.7 with Control module is used in the testing device Shimadzu Autograph AG-X plus 5 kN with video-extensometer for large strain tests such as tensile tests of composite materials with elastomer and viscoelastic materials and a hybrid temperature-humidity chamber WEISS TECHNIK EKE 60.180.70.C with a range from −60 °C to 180 °C.

The testing method for low cycle tests of composites with a textile cord and an elastomer matrix on a test machine with a video-extensometer has these conditions (based on previous research works [18] and according to the mentioned standard [11]): the loading speed 250 mm·min^−1^, the loading speed for a pretest 50 mm·min^−1^, a preload force of 2 N. The initial length between the points for the video-extensometer is 50 mm, which is half of the initial length between the jaws.

The five-cycle loops are used. Every cycle loop consists of five cycles. Every cycle is defined as a load to a certain strain between the jaws of the test machine and an unloading to a certain strain between the jaws. The control of the testing machine was performed by measuring the strain between the jaws with the machine control sensor gauge. The first cycle loop consists of cycles with loading at 30% and unloading at 3% of strain (but not 0% because under approx. 2% of elongation, there is a compressive force during tensile testing for composite specimens with textile reinforcement). The next cycle loop consists of cycles with strain 10% higher than in the previous cycle loop, according to literature [6]. Therefore, the second loop consists of cycles with loading to 40% and unloading to 10% of strain; the third loop consists of five cycles with loading to 50% and unloading to 20% of strain; and the fourth loop has cycles with loading to 60% and unloading to 30% of strain. The fifth loop has cycles with different strains; the loading to 60% is the same as the fourth loop, and the unloading is 5% of the strain. The final step is loading to 100% of the strain. Relaxation stress was not considered [6].

The testing method, as described step by step above, is shown as defined in the control screen panel of the software TrapeziumX in Figure 2. The experiments were carried out at a temperature of 20 °C. The fifth cycle for each cycle loop is evaluated as a result of experiments.

### 2.4. Testing Method for Textile Cords

Special stainless steel pneumatic cord and yarn tensile grips with a force limit of 2 kN and an extended shank are used for the tests of PA66 cord. These pneumatic grips can be used in a hybrid chamber. The load control is performed differently from usual (load control is usually completed by strain measured between the jaws) by strain in the working area of the textile yarn by marked points for the video-extensometer. The preparation of the test specimen for the use of the video-extensometer and special pneumatic cord and yarn tensile grips is completed as described in detail in reference [18]. The initial length between the jaws can be 250 mm or 500 mm. According to reference [19], the initial length between the jaws is chosen at 250 mm, and the distance between the measurement points is 125 mm. The distance is chosen as the maximum possible to obtain the measured strain from the points by the video-extensometer to a strain of about 40% by literature [6]. The real value of this distance is 125.7 ± 1.6 mm. The point on the specimen (the upper point—the upper jaw is movable, the lower jaw is fixed) must be inside the field of view of the video-extensometer during testing.

The authors proposed several methods. The first is simple, based on five cycles without the use of a video-extensometer. The preload was a force of 2.35 N with a preload speed of 10 mm·min^−1^, loading was performed to a strain of 15% (other methods use a strain of 20%), and unloading was performed to a force of 5 N. Subsequently, stress relaxation, or creep, was added. It was also necessary to verify the effect of temperature on stress relaxation and creep.

Based on the results from the first simple method, the authors proposed a modified method for low cycle loading tests of the textile cord that was designed on a test machine with a video-extensometer with three cycle loops and a relaxation time pause at the end of each cycle loop. Every cycle loop consists of five cycles. Every cycle is defined as loading to a certain strain on the video-extensometer of the test machine and unloading to a certain strain on the video-extensometer. The control of the testing machine was performed by measuring the strain on the video-extensometer. The first cycle loop consists of cycles with loading at 5% and unloading at 2.5% of strain. After the fifth cycle, the relaxation time pause is 60 s. The second loop consists of cycles with strain higher by 2.5%: loading to 7.5% and unloading to 5% of strain. After the fifth cycle of the second cycle loop, the relaxation time pause is 60 s. The third loop consists of five cycles that have loading at 10% and unloading at 7.5% of the strain. After the fifth cycle of the third cycle loop, the relaxation time pause is 60 s. The final step is the loading and breaking of the specimen. The loading speed was reduced to 25 mm·min^−1^ because a higher speed is not good for the control by strain on the video-extensometer. The preload was a force of 0.01 N. The testing method, including a hold step for 60 s between each loop, is described in the control screen panel of the software TrapeziumX in Figure 3a. The strain on the video-extensometer is marked on the axis as strain on extens. The experiments were conducted at two temperatures of 20 °C and 120 °C. These temperatures are chosen based on practical requirements to perform the measurement at normal temperatures and at increased temperatures when a change in material parameters is assumed. A graphical representation of the change in strain on the video-extensometer as a function of time is shown in Figure 3b.

## 3. Results and Discussion

### 3.1. Cyclic Tests of Composites

The dependences of true stress on strain on the video-extensometer (and for comparison, true stress on strain between jaws) are in Figure 4, and for fifth cycles, they are in Figure 5, which are result dependences for practical use. These dependencies (Figure 5) can be used as input material data, which describes a textile cap ply for computational simulation of a textile carcass of a sports bike tire with PA66, where it was necessary to solve the material parameters for the simulation of dynamic loading conditions. True stress is defined as the stress obtained by the ratio of the current tensile force to the current cross section of the test specimen obtained based on the change in width measured by the video-extensometer.

The true stress values for the maximum strain of the fifth cycle of the loops are in Table 5. Based on the results (Figure 6), there is a relationship between stress and the number of cycle loops as shown by the second-degree polynomial curve with an R-squared of 1:(1)$$True\;stress_{NoL}\;\left(\mathrm{MPa}\right)=0.0725\times {\left(NoL\right)}^{2}+0.4565\times NoL+1.4725$$
where *NoL*—number of cycle loop.

The relationship can be used to predict the true stress for the next cycle loop under the same testing conditions. For example, if it is necessary to know the stress value for the end of the fifth cycle for the sixth loop of the composite for computational modeling, then the stress value could be 6.8 MPa according to the relationship (1).

Furthermore, it was also necessary to verify the scientific hypothesis and determine whether it is already possible to consider the fourth cycle as stable. The fourth cycle can be considered a stable cycle if the percentage change in the maximum true stress between the fourth and fifth cycles is less than 2%. The percentage changes of maximum true stress between the second and third cycles, between the third and fourth cycles, and between the fourth and fifth cycles are in Table 6.

Based on the results, the fourth cycle can be considered a stable cycle because the percentage change in the maximum true stress between the fourth and fifth cycles is 1.6%.

### 3.2. Cyclic Tests of Textile Cords—Relaxation Stress and Creep

The evaluation will be in terms of the dependence of the force on the strain between the jaws. For textile materials, stress in MPa is not considered, but in e.g., cN·tex^−1^. Therefore, knowledge of force is important from cycle loading tests. Figure 7 shows the dependence on the test without relaxation and the use of the video-extensometer. The method with 15% (preload 2.35 N and 10 mm·min^−1^, the specimen with an initial length of 250 mm between jaws, loading to 15% strain, unloading to 5 N) was used with a loading speed 250 mm·min^−1^. The force is 44.22 N at a strain of 14.94% for the fifth cycle.

Figure 8 shows the results of tests with the same conditions, only with relaxation, stress, and creep for 120 s. The right curve with creep is shifted on the *x*-axis by 4% to the right for better comparison. Relaxation stress (specimen No. 1): force change after a time of 120 s is 3.89 N (force after the fifth cycle is 41.21 N, force after relaxation time is 37.32 N). It changes about 9.4% of force after the fifth cycle. Creep (specimen No. 2): strain change after a time of 120 s is 0.69% (strain after the fifth cycle is 14.9%, strain after relaxation time is 15.59%). It is a change of about 4.6% of strain after the fifth cycle 14.9%.

Force values after individual cycles are in Table 7.

Figure 9 is a relationship between force (for the second to the fifth cycles) and number of cycles as a linear dependence with R-squared of 0.994:(2)$$F_{NoC}\left(N\right)=0.989\times F_{1st^{}}-0.38\times NoC$$
(3)$$e.g.,\;F_{3rd^{}}=0.989\times 43.39-0.38\times 3=41.80\;N$$
where *NoC*—number of cycle loop, *F*_1*st*_—force of the 1st cycle.

Furthermore, the method with 20% strain was changed to compare the results with the method with 15%. A comparison of the dependences between force and strain between jaws is shown in Figure 10. The method is with creep. Force values after individual cycles are in Table 8.

After the fifth cycle: 56.79 N (19.96%) and 43.79 N (14.9%). After a force hold for 120 s: the strain is changed to 20.73% and 15.59%, and the force is 57.16 N for the strain of 20.73%. The strain changes are 0.77% (method with 20%) and 0.69% (method with 15%).

Based on the results, there is a relationship between force and number of cycles: (4)$$F_{NoC}\left(N\right)=0.989\times F_{1st^{}}-0.51\times NoC$$
(5)$$e.g.,\;F_{4th^{}}=0.989\times 60.37-0.51\times 4=57.67\;N$$

Thus, the number 0.51 in the formula for the 20% method is different from that for 15% (the value in the formula was 0.38). Here, it is interesting that if for the 15% method 15 is divided by the coefficient 0.38 = the result is 39.47. A similar result is obtained for the 20% method, i.e., 20% divided by 0.51 = the result is 39.22. So, based on this, it can be deduced that if the experiment was carried out for another method, e.g., 25% method, then the formula would be the same, only the coefficient of the *NoC* multiplier would be approx. 0.63 (it is 25/39.47).

### 3.3. Cyclic Tests of Textile Cords—Temperature Influences

The test results as force-strain on video-extensometer dependences by method with three cycle loops for two temperatures (the temperature chamber was used) are in Figure 11. The relaxation time was in each cycle loop. The dependencies can be used as input material data for textile cords in the computational simulations of tire casings using a custom material model definition. Software tools used for computational simulation will typically allow for the definition of custom material models.

The force values after relaxation time for each cycle loop are in Table 9. The transformation of force values to stress values is given in Table 9. The fineness (linear density) of the textile-conditioned cord is 944 dtex; therefore, stress can be obtained as force divided fitness. The unit of stress is deliberately adjusted so that the calculated stress values are in the range from 0 to 100 for better comparison of the values with each other. Therefore, the stress for textile materials is given in multiples of the force unit, e.g., in cN (100 cN is 1 N), divided by fineness, e.g., in tex instead of dtext (10 tex is 1 dtex). In our case, we chose the unit of stress as cN·tex^−1^.

The force-time dependences are shown in Figure 12. Tenacities (the tensile stress expressed as force per unit fineness) for nine specimens are 75.99 ± 1.84 cN·tex^−1^ for a temperature of 20 °C and 37.22 ± 2.85 cN·tex^−1^ for a temperature of 120 °C, and the ductility (strains at break) are 17.79 ± 0.5% for a temperature of 20 °C and 12.11 ± 0.52% for a temperature of 120 °C. The values for a temperature of 20 °C are different from cord producer values [10], but they are closer to the data from the patent [20], which deals with the cord construction PA66 470 × 2 (it means fineness is 940 dtex). The tenacity for a temperature of 20 °C reported in literature [13] is 86.2 cN·tex^−1^, and the ductility (referred to in this literature as elongation at break) is 18.6%. The difference in measured tenacity from the manufacturer’s published [13] is 11.8%, and the difference in ductility is 4.4%. Based on patent [20], tenacity obtained from a breaking force of 73 N is 77.66 cN·tex^−1^, and ductility is 18%.

## 4. Conclusions

New methods are presented as possible test references for studying composites and textile cords. On the basis of the results, the fourth cycle in every cycle loop can be considered a stable cycle for polymer composites with PA66 cords because the change in the maximum true stress between the fourth and fifth cycles is 1.6%. The method for textile cords considers stress relaxation.

A simple and fast relationship is found that describes the value of the force at each end of the cycles for a textile cord. It can be used to predict forces for specific strains and can be used for practical use. These preliminary results point to the possibility of further research into textile material properties. The results of the polymer composite and textile cord tests provide a better understanding of the mechanical properties under cyclic loading.

The paper evaluated the effect of temperatures on the material properties of textile cord and the relaxation effect, which is important for the assessment of practical applications. Knowledge of this effect is important to assess the change in material parameters that occurs during the vulcanization process of tire casings.

The obtained equations can be used to determine input data for computational simulation. It would be appropriate for further research to apply a video-extensometer for online monitoring and recording of deformations on the surfaces of test specimens during cyclic loading tests. For composite specimens, the authors recommend also performing biaxial cyclic load tests that would be even closer to real load conditions.

## Figures and Tables

**Figure 1 polymers-15-02358-f001:**
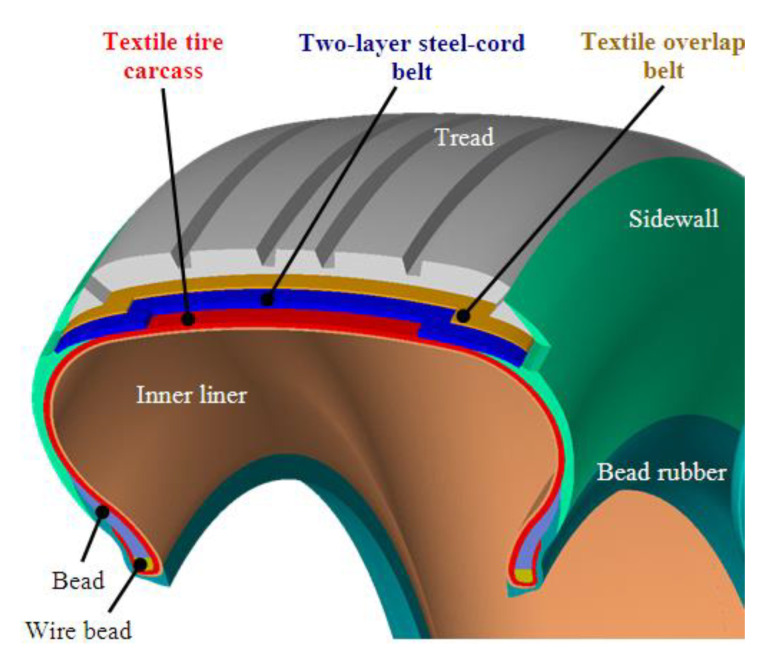
Construction of radial tire casing [6].

**Figure 2 polymers-15-02358-f002:**
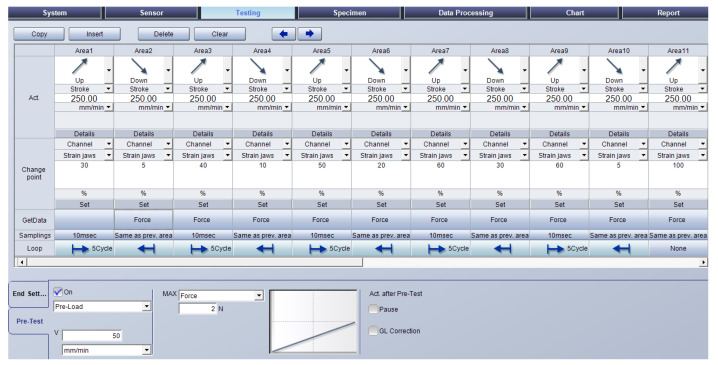
Testing method for composite specimens in software TrapeziumX.

**Figure 3 polymers-15-02358-f003:**
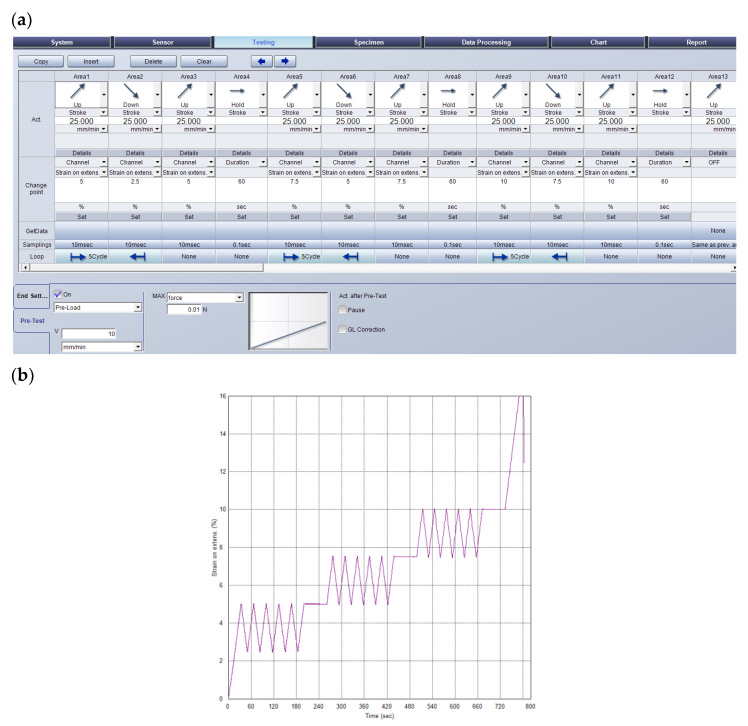
Testing method for textile cord with three cycle loops and relaxation time in software TrapeziumX (**a**) and change in strain on video-extensometer (marked as strain on extens.) as a function of time (**b**).

**Figure 4 polymers-15-02358-f004:**
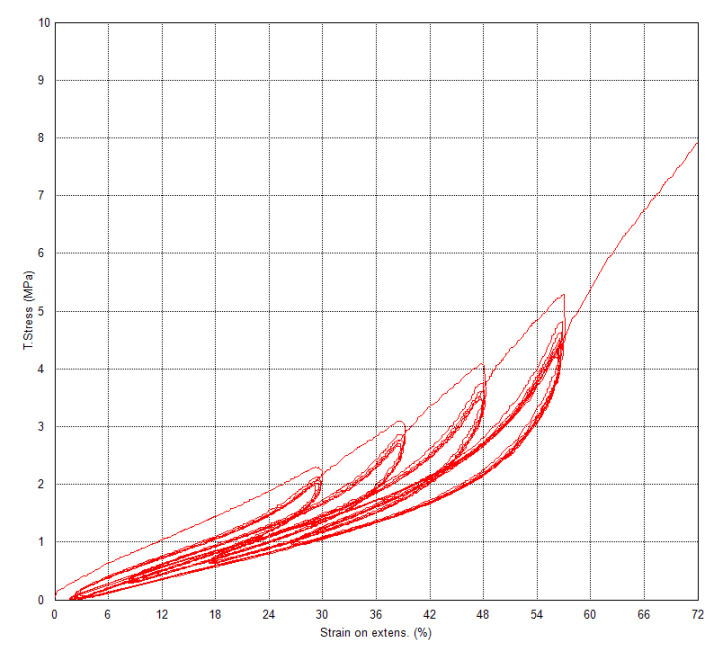
Composite—dependences of true stress on strain on video-extensometer.

**Figure 5 polymers-15-02358-f005:**
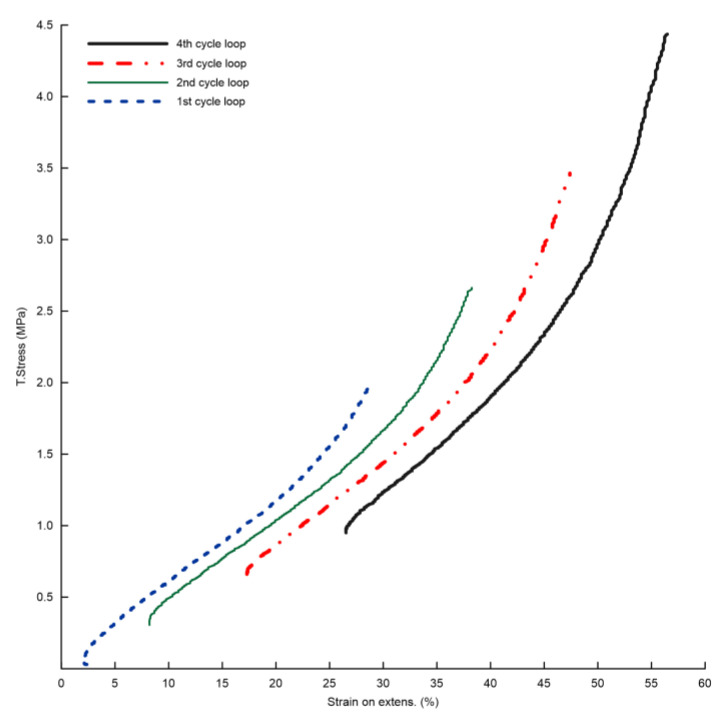
Composite—dependences of true stress on strain on video-extensometer for fifth cycles.

**Figure 6 polymers-15-02358-f006:**
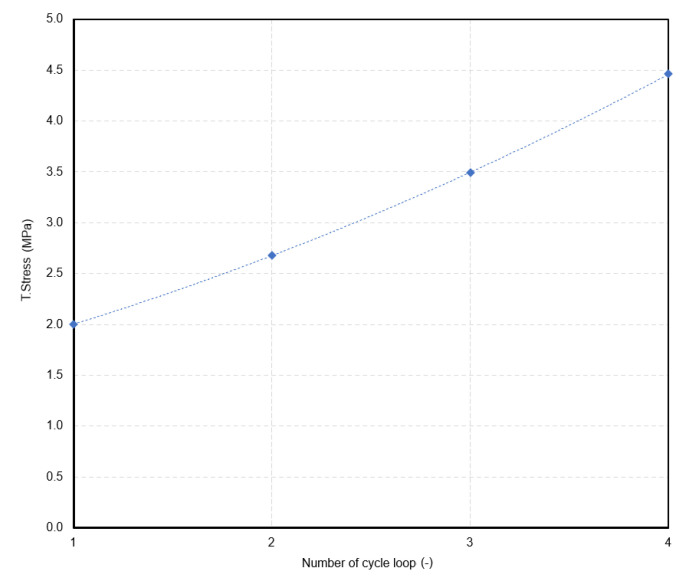
Composite—polynomial curve between true stress at the end of the fifth cycle and number of cycle loop.

**Figure 7 polymers-15-02358-f007:**
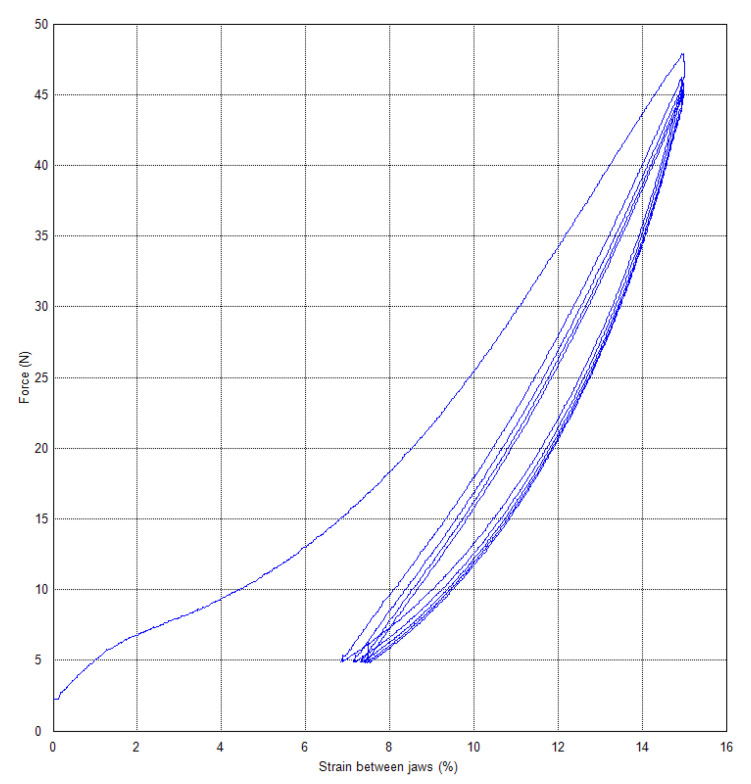
PA66—cyclic loading without relaxation stress.

**Figure 8 polymers-15-02358-f008:**
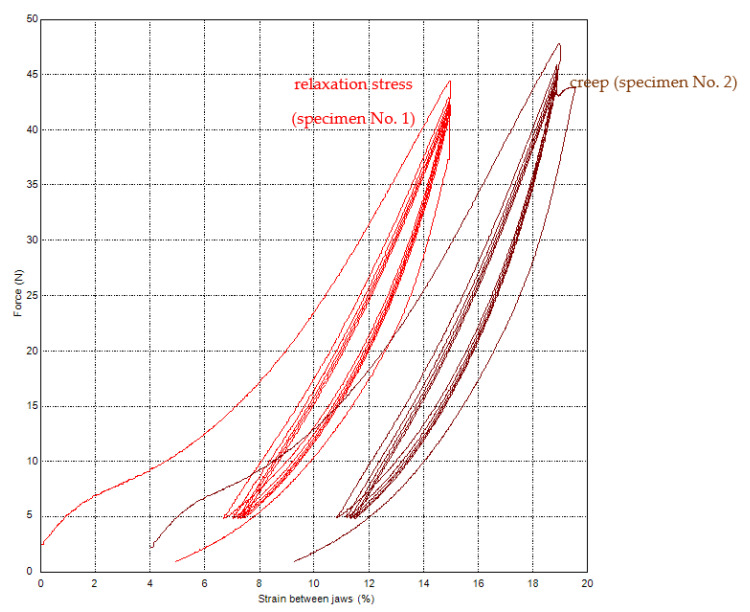
PA66—cyclic loading with relaxation stress (specimen No. 1) and with creep (specimen No. 2).

**Figure 9 polymers-15-02358-f009:**
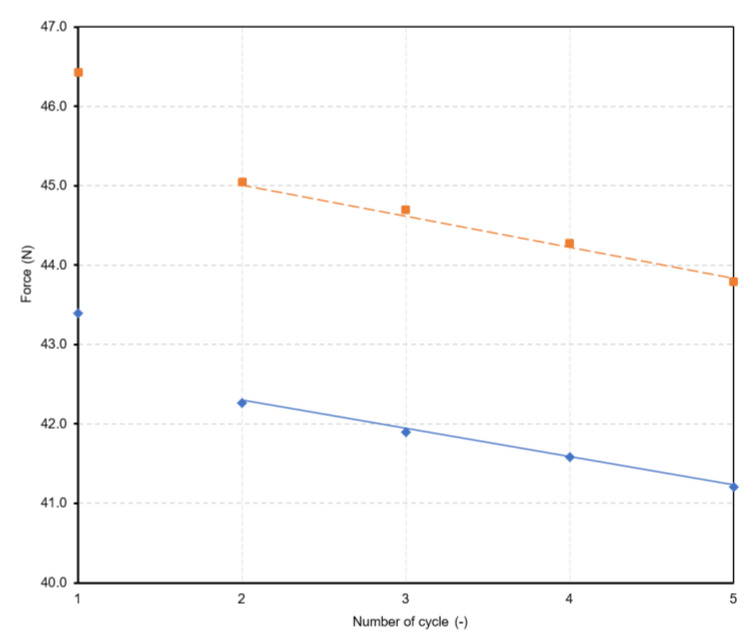
Linear dependences between force and number of cycle (red line for creep, blue line for relaxation).

**Figure 10 polymers-15-02358-f010:**
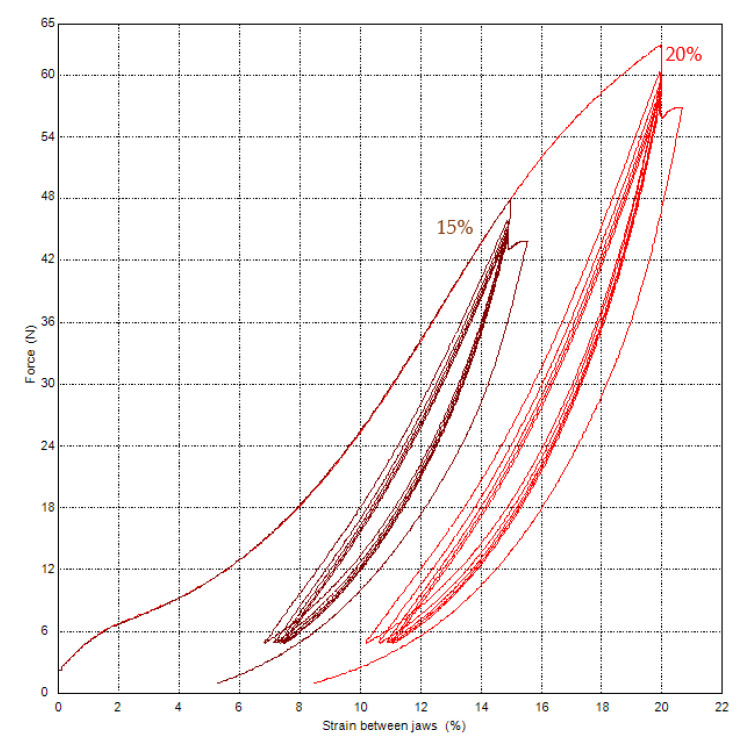
PA66—cyclic loading with creep—comparison of 15% and 20%.

**Figure 11 polymers-15-02358-f011:**
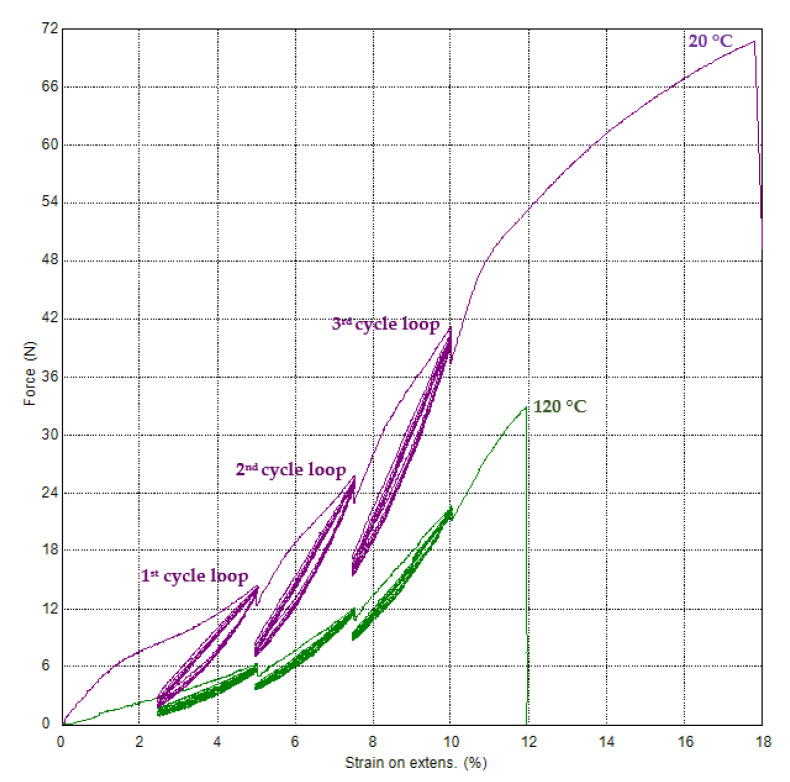
PA66—tensile force–strain on video-extensometer dependences for three cycle loops with relaxation for two temperatures.

**Figure 12 polymers-15-02358-f012:**
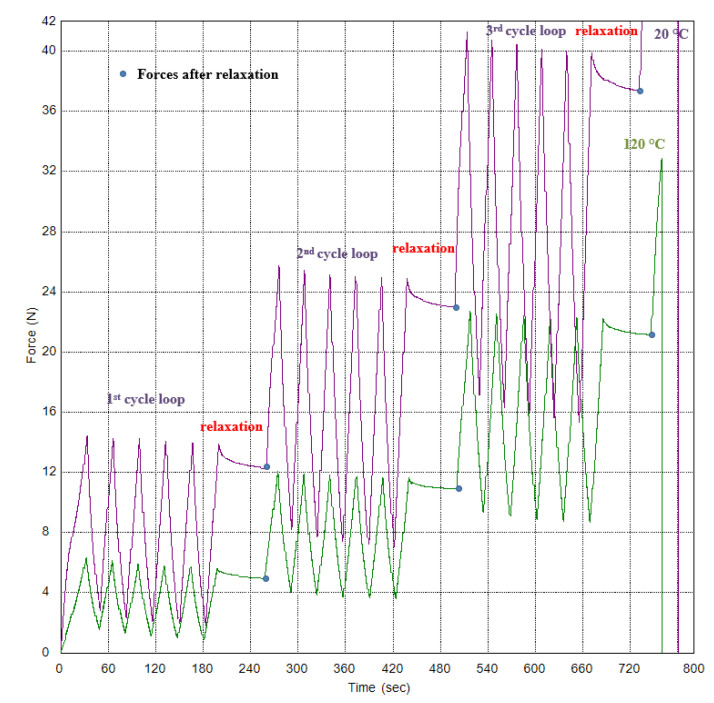
PA66—tensile force–time dependences for three cycle loops with relaxation for two temperatures.

**Table 1 polymers-15-02358-t001:** Materials of textile cords for tire casings.

Material	Application
polyester (PES)	carcass
polyamide 6(PA6)	overlap belt
polyamide 66 (PA66)	overlap belt
rayon	carcass
aramid	carcass for sport purposes

**Table 2 polymers-15-02358-t002:** Parameters of Dunlop 215/40 R17 87V Extra load SS SPORT MAXX tire casing.

Construction Parts of Tire Casing	Steel-Cord Belt	Textile Overlap Belt	Textile Carcass
number of layers	2	1 and 2	2
thickness of one layer, mm	1.38–1.44	1.10 and 1.75	1.40–1.50
diameter of cord, mm	0.80	0.44	0.50
EPM, m^−1^	620–680	1400	no data

**Table 3 polymers-15-02358-t003:** Parameters of Continental 245/40 R18 97Y Extra load ContiSportContact 3 tire casing.

Construction Parts of Tire Casing	Steel-Cord Belt	Textile Overlap Belt	Textile Carcass
number of layers	2	1	2
thickness of one layer, mm	0.97	1.20	0.92
diameter of cord, mm	0.60	0.69	0.51
EPM, m^−1^	920–960	775	no data

**Table 4 polymers-15-02358-t004:** Material parameters of Kordsa T-728.

Material Parameter	Value	Publication Source
breaking force	81.4 N	[15]
breaking strength	8.3 kg	[13]
tenacity	86.2 cN·tex^−1^	[13]
strain at break (ductility)	18.6%	[13,15]
strain (elongation) at 45 N set force	9.6%	[15]
elongation at 4.5 kg set force:	9.6%	[13]
number of filaments	140	[15]
%shrinkage lenzing 177 °C for 2 min	6.6%	[13,15]

**Table 5 polymers-15-02358-t005:** True stress at the end of the fifth cycle for each cycle loop of composite.

Number of Cycle Loop	True Stress, MPa
1st	2.00
2nd	2.68
3rd	3.49
4th	4.46

**Table 6 polymers-15-02358-t006:** True stress at the end of the fifth cycle for each cycle loop of composite.

Number of Cycle Loop	% Change of Maximum True Stress between the 2nd and 3rd Cycles,	% Change of Maximum True Stress between the 3rd and 4th Cycles,	% Change of Maximum True Stress between the 4th and 5th Cycles,
1st	3.3	2.2	1.6
2nd	3.3	2.0	1.6
3rd	3.4	2.5	1.6
4th	3.8	2.4	1.5

**Table 7 polymers-15-02358-t007:** Method to strain of 15%—forces after individual cycles for strain of 15%.

Number of Cycle	Force—Specimen No. 1, N	Force—Specimen No. 2, N
1st	43.39	46.43
2nd	42.26	45.05
3rd	41.90	44.70
4th	41.58	44.28
5th	41.21	43.79

**Table 8 polymers-15-02358-t008:** Method to strain of 20%—forces after individual cycles for strain of 20%.

Number of Cycle Loop	Force, N
1st	60.37
2nd	59.04
3rd	58.40
4th	57.49
5th	56.79

**Table 9 polymers-15-02358-t009:** Forces after relaxation time for each cycle loop.

Number of Cycle Loop	Force for 20 °C, N	Force for 120 °C, N	Stress for 20 °C, cN·tex^−1^	Stress for 120 °C, cN·tex^−1^
1st (strain 5%)	12.25	4.95	12.97	5.23
2nd (strain 7.5%)	22.95	10.87	24.31	11.51
3rd (strain 10%)	37.33	21.11	39.54	22.36

## Data Availability

The data presented in this study are available on request from the corresponding author.

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
