# Peer review of "Cyclic Testing of Polymer Composites and Textile Cords for Tires"

_polymers, 2023, doi:10.3390/polym15102358_

Round 1

Reviewer 1 Report

This paper deals with the specific testing of polymer composites and textile cords used as reinforcement for the composites. Textile cord-rubber composites were selected for the work. Generally, the use of lighter and cheaper reinforcers with high strength and durability is in great demand. Thus, the performed research is of interest and can be recommended for publication in Polymers. However, before that, the following comments can help improve the manuscript,

*Please change the title into:

“Specific Cyclic Testing of Polymer Composites and Textile Cords: Design of Experimental Methods for Determination of Input Data for Computational Simulation”.

*It is mentioned that: “A low uniaxial cyclic tensile load was carried out for two temperatures of 20 °C and 120 °C”.

What is the reason for choosing these two temperatures? An explanation is required.

*English needs revision. There are many sentences that are hard to read. For example,

“This makes it possible to experimentally the load that occurs during operation.”

And many more. Please double-check the whole text.

*The most important new result of the experimental works can be mentioned at the end of the abstract.

*What is the criteria to select cord angles of 0°, 25°, 45°, 60° and 90°?

*The experimental explanations should be presented together with the related pictures. Please provide it for section 2.

*It is better to have one kind of format for figures. Figures 4, 5, and 6 are in different styles and formats.

*All the numerical data presented in Tables shall have the same decimal. For example, write down 41.90 and 41.70 in Table 5.

There are some sentences that have readability problems.

Author Response

The authors of the article are sincerely grateful to the dear reviewer for carefully reading of the manuscript and for valuable suggestions and comments.

We have made the changes in the manuscript in yellow text. We edited the title, changed the abstract, and more. We have corrected the figures according to your recommendation.

With respect and best regards,

the team of authors of the article.

Reviewer 2 Report

The paper on Specific Cyclic Testing of Composites and Textile Cords by J. Krmela  et al., presents a collection of results of simple uniaxial testing of very specific materials. 

The paper is interesting because of the purpose of analyzing how to make adequate tests on the proposed materials to obtain the right material characterization that could be used into numerical models to analyze complex designs. Unfortunately, the submitted paper is quite unsatisfactory: the writing quality is poor, with many repetitions, and many loose explanations and missing information. An important criticism is that the Authors refer to the new methods without any justification or reasons. There is no sound discussion or proposal about the selection and definition of the methods. 

The Reviewer would be glad to review a thoroughly revised and improved version of the paper. The results, the purpose of the paper and the developed testing methods are of general interest if soundly presented, but hardly as it is presented now.  

Neither the Authors or the Editor provided a line-numbered version. Please, consider that for future revisions. The draft is sent back marked and comments follow in a list.

The quality of the text is limited. I will not make such a Review. It is very time consuming and, moreover, beyond my duties as Reviewer. Anyway, many comments are included in the Review that would help in this revision.

Author Response

The authors of the article are sincerely grateful to the dear reviewer for carefully reading of the manuscript and for valuable suggestions and comments.

We have made the changes in the manuscript in green text. We edited the title, changed the abstract, and more. We have corrected the figures according to your recommendation.

With respect and best regards,

the team of authors of the article.

Reviewer 3 Report

The reviewed article on cyclic tests of composites and textile cords based on PA66 fiber refers to the determination of experimental material characteristics of the necessary input data for the appropriate software simulating the fatigue life of a given product. This manuscript in this form cannot be published. The idea of this article needs to be rebuilt.

The first fundamental remark: how was the correctness of the presented methodology for determining material input data to the proposed software verified?

The reliability of estimating specific values using numerical methods requires their validation experimentally. Where do the authors of the manuscript expose this?

There is no assessment of the state of knowledge in the introduction.

What is the novelty of the adopted goal?

In section 2.2. The described research material might be better presented in a table. It is difficult for the reader to find out what parameters of the analyzed composite material were adopted.

Sections 2.2 and 2.3 such names are a bit of a misunderstanding.

I think it's about developing methodologies for identifying material data necessary for proprietary software and this should be reworded.

Section 3 describes the identification of material constants.

In this section, it is appropriate to describe the concept of this identification with a mathematical approach. The two graphs presented (Fig. 2 and 3) what do they contribute to the identification of material constants? Are you sure formula (1) describes a polynomial of the third degree?

Results presented in section 3.2: no discussion. How to read the values listed in table 5 with the graph (Fig. 7)? Maybe it would be worth to enlarge the regions of maximum force values.

Analyzing the results from table 6 and the description right below the table, the value is inconsistent with the 5th cycle. Only the next sentence makes the reader aware of these values. The description of table 6 is imprecise.

What the authors meant: "Based on the results, there is a relationship between force". So this relationship is between force and (?)

How to read the results from table 7 and the graph (Fig. 10)? The results do not correspond to each other (see 2nd cycle loop), what about the declared strains (%)?

What does Fig. 11 do? What can the reader conclude from it? Where is the comparison, for example, with analytically estimated results and showing that the proposed methodology works? How does this relate to the results of other researchers? If they are comparable to the manufacturer's results and work [17], where is it shown and what error occurs? Which values are more reliable those from [9] or [17] or presented in this manuscript?

The authors' statement that "It can be used to predict forces and can be used for practical use": there is no verification and you should be careful here.

In conclusion, the work is a research report and not a scientific study.

Author Response

The authors of the article are sincerely grateful to the dear reviewer for carefully reading the manuscript and for valuable suggestions and comments.

We have made the changes in the manuscript, not only in blue text. Some of your comments are similar to what other reviewers have had. We edited the title, changed the abstract, and more. 

With respect and best regards,

the team of authors of the article.

Round 2

Reviewer 2 Report

The Reviewer acknowledges the thorough revised version provided by the Authors. Many comments have been considered to improve the descriptions and complete the missing information. I still recommend that the Authors consider the following comments to improve the information or style about few points 

The Authors have made an effort to improve the quality. A new reading and check will help. 

Author Response

The authors of the article would like to thank the reviewer for comments. 

We have made the changes in the manuscript in red text

"Please, consider including a picture of the composite specimens. Mention the connection of the specimens in relation of Tables 2 and 3 parameters. " - we did not insert a picture of the specimens in the article, because the specimens are a "black rubber rectangles" with a dimension of 195 x 35 mm. Thank you for  understanding.

Have a nice day.

Best regards the team of authors of the article.

Reviewer 3 Report

The authors took into account the comments and made appropriate corrections.

Author Response

The authors of the article would like to thank the reviewer.

Have a nice day.

Best regards the team of authors of the article.